# Independent somatic evolution underlies clustered neuroendocrine tumors in the human small intestine

Erik Elias [1,2,6], Arman Ardalan[3,6], Markus Lindberg[3], Susanne E. Reinsbach[3], Andreas Muth [1,2], Ola Nilsson[4,5], Yvonne Arvidsson[5] & Erik Larsson [3✉]

Small intestine neuroendocrine tumor (SI-NET), the most common cancer of the small bowel, often displays a curious multifocal phenotype with several tumors clustered together in a limited intestinal segment. SI-NET also shows an unusual absence of driver mutations explaining tumor initiation and metastatic spread. The evolutionary trajectories that underlie multifocal SI-NET lesions could provide insight into the underlying tumor biology, but this question remains unresolved. Here, we determine the complete genome sequences of 61 tumors and metastases from 11 patients with multifocal SI-NET, allowing for elucidation of phylogenetic relationships between tumors within single patients. Intra-individual comparisons revealed a lack of shared somatic single-nucleotide variants among the sampled intestinal lesions, supporting an independent clonal origin. Furthermore, in three of the patients, two independent tumors had metastasized. We conclude that primary multifocal SI-NETs generally arise from clonally independent cells, suggesting a contribution from a cancer-priming local factor.

[1] Endocrine and Sarcoma Surgery, Department of Surgery, Institute of Clinical Sciences, Sahlgrenska Academy at University of Gothenburg, Gothenburg, Sweden. [2] Department of Endocrine and Sarcoma Surgery, Surgical Clinic SU/S, Sahlgrenska University Hospital, Gothenburg, Sweden. [3] Department of Medical Biochemistry and Cell Biology, Institute of Biomedicine, Sahlgrenska Academy at University of Gothenburg, Gothenburg, Sweden. [4] Department of Laboratory medicine, Institute of Biomedicine, Sahlgrenska Cancer Center, Sahlgrenska Academy at University of Gothenburg, Gothenburg, Sweden. [5] Department of Pathology, Sahlgrenska University Hospital, Gothenburg, Sweden. [6]These authors contributed equally: Erik Elias, Arman Ardalan. ✉email: erik.larsson@gu.se

Cancer is considered to be an evolutionary process involving cycles of random oncogenic mutations, selection, and clonal expansion[1]. In most malignant epithelial tumors (carcinomas), a single primary tumor will typically form, followed by metastases arising as cells disperse from the tumor. More rarely, carcinomas instead display multifocality, i.e., multiple separate tumors at the primary site, which may be due to predisposing germline variants, local mutagenic exposure, or localized spread or expansion of cancer-primed mutated clones[2,3].

Of particular interest are small intestine neuroendocrine tumors (SI-NETs), the most common cancer of the small intestine, where ~50% of cases display a striking multifocal phenotype that often involves 10 or more morphologically identical tumors clustered within a limited intestinal segment, commonly centered around a regional lymph node metastasis[4,5]. SI-NET has a reported incidence of ~1.2 per 100,000[5] and often presents with distant metastases at diagnosis, thus precluding curative treatment[4,6]. However, the somatic mutational burden of SI-NET is low, and there is an unusual absence of somatic driver mutations[7–9]. Consequently, underlying tumorigenic mechanisms are poorly understood and actionable genetic drug targets are lacking.

The evolutionary relationships that govern multiple intestinal tumors and metastases in SI-NET can give insight into the underlying biology, but earlier efforts to determine this have yielded conflicting results[10,11]. A study of copy number alterations (CNA) in multifocal SI-NET primary tumors found that loss of chromosome 18 (chr18), the most frequent CNA, can affect different chromosome homologs in different samples within a patient[12], compatible with an independent clonal origin or late loss of chr18. Alternatively, multifocal SI-NETs have been proposed to represent drop metastases originating from regional lymph nodes, consistent with the limited spatial distribution of the intestinal tumors[13]. Adding to the challenge is the low mutational burden, which reduces the number of exonic genetic markers.

In this work, we perform whole genome sequencing (WGS) on 61 separate intestinal tumors and adjacent metastases from 11 patients, enabling us to conclusively determine the evolutionary trajectory of multifocal SI-NET lesions within single individuals. We conclude that primary multifocal SI-NETs normally arise from clonally independent precursors. Furthermore, despite clonal independence, we find that more than one intestinal lesion can give rise to metastases within a single SI-NET patient.

## Results

**Evolutionary trajectories in multifocal SI-NET.** We initially performed whole genome sequencing (WGS) on six intestinal tumors, three adjacent lymph node metastases, two peritoneal metastases, and a normal blood sample from a patient with suspected SI-NET (Patient 1) that underwent surgery with curative intent (Fig. 1a). All lesions showed typical SI-NET morphology and stained positively for established diagnostic SI-NET markers (Supplementary Fig. 1). Samples were sequenced at an average coverage of 34.5–43.1× (Supplementary Data 1). This was followed by somatic mutation calling using strict filters including rigorous population variant removal to avoid false positives in phylogenetic analyses. Resulting genome-wide single nucleotide variant (SNV) burdens varied from 353 to 1,749 (0.13–0.62 per Mb; Supplementary Data 1).

We next determined pairwise shared somatic SNVs between samples. Surprisingly, a common set of 667 SNVs was shared between a single primary tumor (denoted C) and all five metastases (G–K), while overlapping mutations were essentially lacking between the other samples (Fig. 1b). The results were thus not compatible with a common clonal origin for the intestinal tumors, expected to result in hundreds to thousands of shared mutations genome-wide in the case of late-onset cancer. Instead, phylogenetic analysis supported that the intestinal tumors had developed independently, with a single, centrally located tumor metastasizing first to the lymph nodes and then further to the peritoneum (Fig. 1b, c).

To validate the initial findings, we analyzed 50 additional tumors from ten SI-NET patients, plus matching normal blood samples, using WGS at 29.8–45.0× coverage (Patients 2–11; Fig. 2; Supplementary Data 1). The bulk of the material (Patients 3–11) consisted of previously sampled tumors stored at a local biobank. Between three and 11 intestinal tumors and at least one lymph node metastasis was sequenced for each patient, and three cases included a liver metastasis. Macroscopically normal small intestinal mucosa samples were additionally included in four cases. Samples were selected to have sufficient tumor material and purity, and all tumor samples stained positively for SI-NET markers (Supplementary Figs. 2–5). One tumor had a lower-than-expected mutational burden (143 SNVs), explained by low sample purity, while others varied between 411 and 3,390 SNVs (0.15–1.21 per Mb; Supplementary Data 1). In comparison, between 11 and 40 SNVs were called in the normal mucosa samples, where widespread clonal somatic mutations are not expected, supporting that somatic mutation calls had high specificity.

Mirroring the result from Patient 1, shared somatic mutations were essentially absent in between intestinal tumors in all ten additional patients (Fig. 2). In contrast, strong SNV overlaps were seen between individual metastases and specific intestinal tumors, ranging from 82 to 2053 and above 290 SNVs in all pairs but one. For example, in Patient 2, a single intestinal tumor (G) out of 11 that were sampled showed a striking relatedness to an adjacent lymph node metastasis (L; 1,214 shared SNVs; Fig. 2b). Similar to Patient 1, the metastatic primary tumor was centrally located in the resected section (Fig. 2a, c). Detailed positional data was lacking for the remaining patient samples, as these were derived from archival material.

While individual metastases showed credible overlaps only with a single primary tumor, in three cases (Patients 6, 7, and 10) we found that two different primary tumors had metastasized, resulting in independent lymph node or liver metastases (Fig. 2g, h, k). Given the lack of a common evolutionary trajectory for the primary tumors, this supports that acquisition of metastatic properties is not a rare event in multifocal SI-NET. In Patient 8, the sample phylogeny suggested two independent metastatic events from a single primary tumor (A), with an earlier event giving rise to a liver metastasis (D) and a later event resulting in a lymph node metastasis (C; Fig. 2i). Shared SNVs typically had higher variant allele frequencies (VAFs) than other variants, in both primaries and metastases, consistent with continued somatic evolution and subclonal expansions following the metastatic events (Supplementary Fig. 6). In a single case (Patient 9), a lymph node metastasis could not be associated to any of two available intestinal tumors, likely explained by an incomplete sampling of intestinal lesions (Fig. 2j).

False shared somatic SNVs may arise, for example, due to failure to detect germline SNPs at specific positions in the blood normal, since this data is common to all samples in a patient. However, only a small number of additional common SNVs were observed beyond the main genetic relationships (Fig. 1b and Fig. 2b, d-l), some of which could be dismissed as false positives by manual inspection (Supplementary Data 2). In Patient 5, one additional tumor (D) shared eight credible SNVs with the metastasis (F) (Fig. 2f). This overlap grew to 66 when high-confidence variants were whitelisted for relaxed calling in other

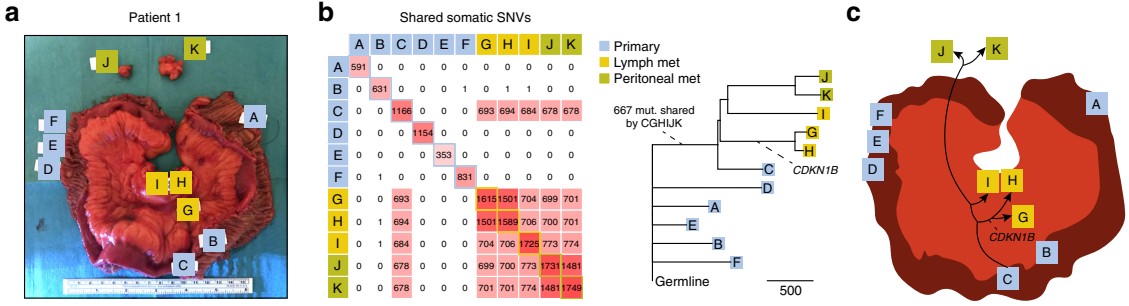

**Fig. 1 Whole genome sequencing of 11 primary tumors and metastases from a single SI-NET patient supports independent clonal evolution. a** Section of the small intestine from a patient (Patient 1) harboring six primary tumors (A–F), three lymph node metastases (G–I), and two peritoneal metastases (J-K). **b** Pairwise analysis of shared somatic SNVs based on whole genome sequencing. Primary tumor C and the five metastases shared a common set of 667 mutations. A maximum parsimony phylogenetic tree is shown, with the number of SNVs in each branch given by the scale marker. An indel in *CDKN1B*, a known driver event, is indicated. Bootstrap support was > = 98% for all major branches. **c** Proposed model, where all metastases originate from a single primary tumor, and where all primaries are unrelated in terms of somatic evolution. Met, metastasis; SNV, single nucleotide variant. Source data are provided as a Source data file.

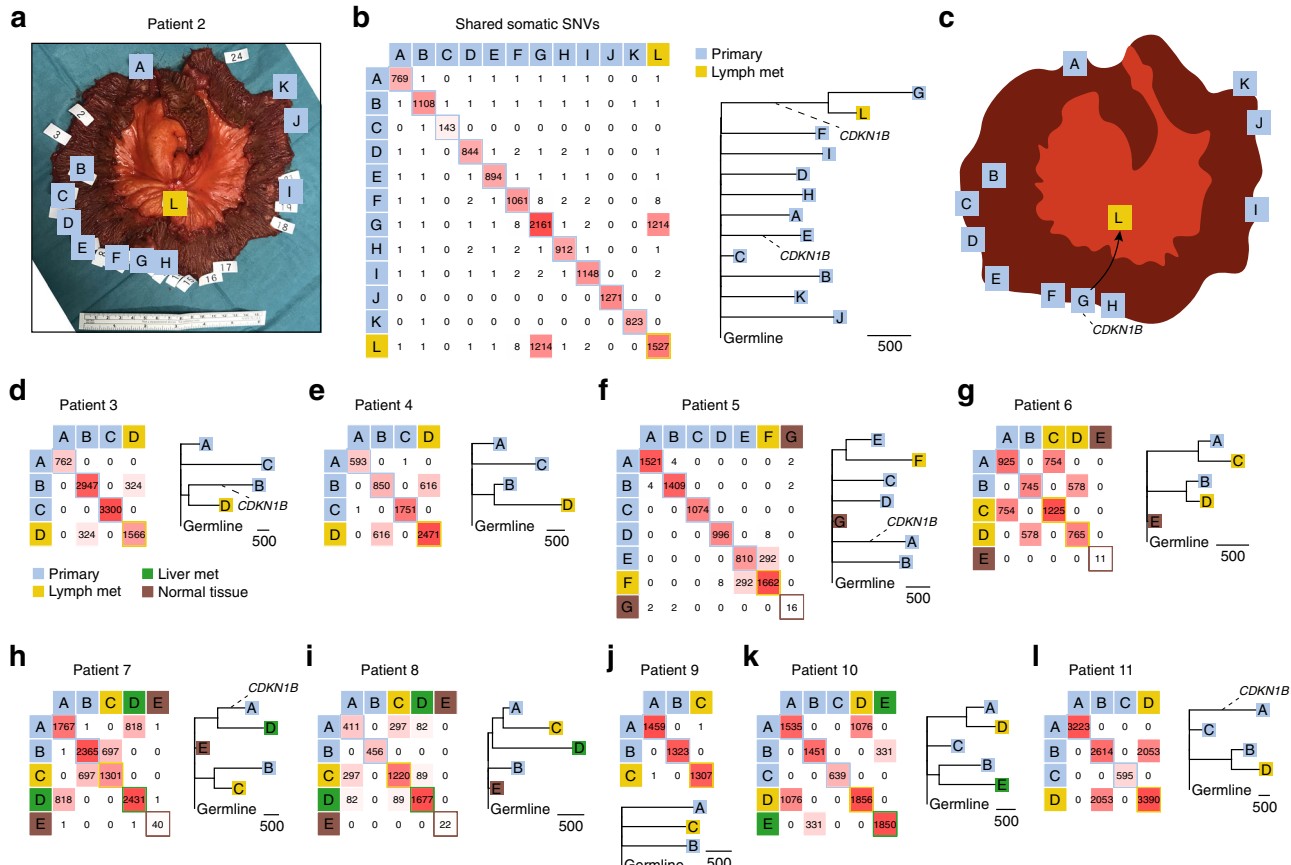

**Fig. 2 Whole genome sequencing of 50 primary tumors and metastases from 10 additional SI-NET patients confirms independent clonal origin. a** Section of the small intestine from a patient (Patient 2) harboring 11 primary tumors (A–K) and one lymph node metastasis (L). White labels indicate all identified tumors while letters indicate sequenced samples with sufficient purity and tumor material. **b** Pairwise analysis of shared somatic SNVs in Patient 2. Primary tumor G and the metastasis shared 1,214 mutations. The known *CDKN1B* driver event is indicated in the phylogenetic tree (the number of SNVs in each branch is given by the scale marker). **c** Proposed model. **d–l** Similar to panel **b**, based on archival material from nine additional patients that also included liver metastases and normal mucosa samples. Bootstrap support was 100% for all major branches in the phylogenetic trees. Met metastasis, SNV single nucleotide variant. Source data are provided as a Source data file.

samples, allowing more sensitive detection of subclonal shared SNVs (Supplementary Fig. 7). These variants were present as low-VAF traces in the metastasis while having normal VAFs in the tumor, compatible with contamination during sample handling or, possibly, metastatic spread to the same lymph node from two tumors with a minor contribution from D (Supplementary Fig. 8). 13 low-VAF variants shared between an intestinal tumor (A) and a lymph node metastasis (C) were also uncovered in Patient 9, in addition to a single high-confidence variant called in both samples (Supplementary Fig. 9). In Patient 2, eight SNVs were

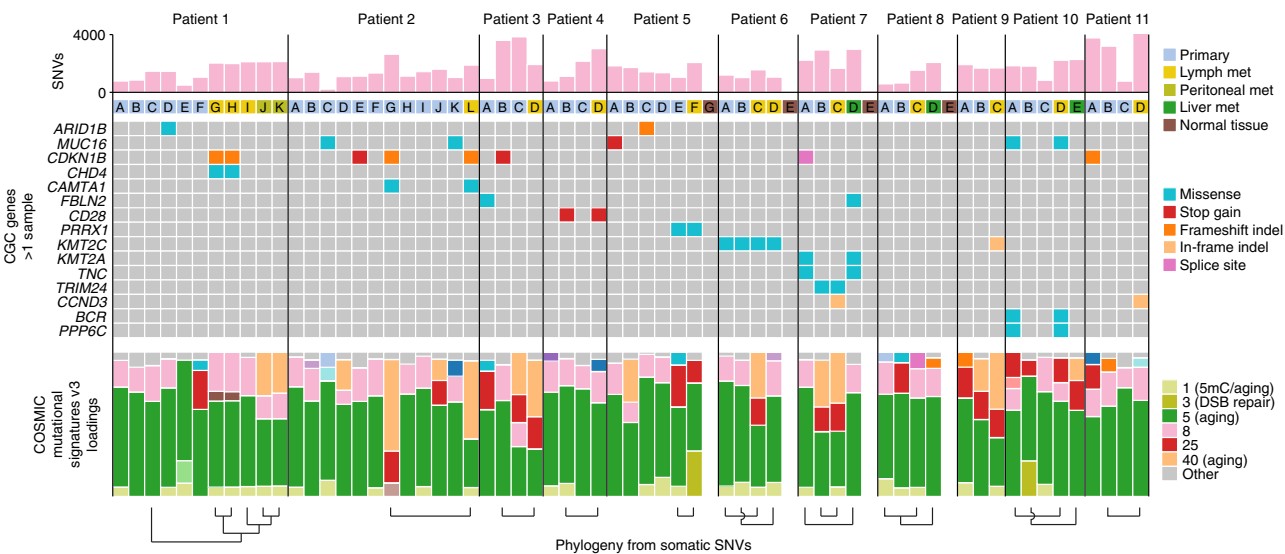

**Fig. 3 Overview of potential driver mutations and mutational signatures for all 11 patients.** All Cancer Gene Census (CGC) genes with non-synonymous mutations in more than one sample are shown. Mutational signature loadings, i.e., relative contributions, refer to COSMIC v3 definitions[16] and were dominated by the closely related signatures SBS5 and SBS40 (see "Supplementary Fig. 16" for a full signature legend). Phylogenetic relationships previously inferred from somatic SNVs (Figs. 1, 2) are indicated at the bottom. Somatic SNV burdens, potential driver mutations and mutational signature loadings in this figure were determined using less rigorous population variant filtering compared to the phylogenetic analyses (see "Methods"). CGC Cancer Gene Census, met metastasis, SNV single nucleotide variant, 5mC 5-methylcytosine, DSB double strand break. Source data are provided as a Source data file.

shared between the main metastatic primary tumor (G), an adjacent tumor (F, 5 mm apart, Supplementary Fig. 10) and the metastasis (L) (Fig. 2b). All had normal VAF distributions in all three samples arguing against contamination (Supplementary Fig. 11) and all but one passed manual inspection (Supplementary Data 2). Overlaps of this size are too small to represent a common clonal origin in a late-onset cancer, and are likely explained by lineage-specific somatic mutations established early during organismal development[14].

**Driver mutations and mutational signatures.** The median age of the patients was 76 years, ranging from 68 to 81 (Supplementary Table 1). The observed burdens and number of pairwise overlapping variants are thus roughly on par with a mutation rate similar to healthy human neurons, estimated to accumulate ~23–40 SNVs/year[15] (Fig. 3). Analysis of mutational signatures indeed supported major contributions from COSMIC Signature SBS5, a ubiquitous aging-associated mutational process, or SBS40 which is closely related to SBS5[16] (Fig. 3). The observed variability in signature loadings may in part be methodological, since the trinucleotide substitution profiles of the samples were in practice highly similar across the cohort (Supplementary Fig. 12). Within-patient burden variability was to a large degree explained by variable sample purity (Supplementary Fig. 13). These results confirm the mutationally quiet nature of SI-NET.

Analysis of potential somatic driver mutations in coding genes mirrored earlier published results, with infrequent variants in *CDKN1B* emerging as the main recurrent event (eight samples, six independent events; Fig. 3; Supplementary Data 3)[7]. In Patient 1, the same *CDKN1B* frameshift indel occurred in two of the metastases (G, H) in a pattern consistent with the inferred phylogeny (Fig. 1b, c). In Patient 2, the same *CDKN1B* frameshift indel was found in the metastatic primary (G) and the metastasis (L), thus again consistent with the phylogeny, while one non-metastatic primary carried a *CDKN1B* stop gain variant (Fig. 2b, c). Patients 3 and 7 carried *CDKN1B* variants (stop gain and splice donor) in metastatic primary tumors that were missing in the corresponding metastases (Fig. 2d, h). These variants were

among a large number of private variants present at relatively low VAF in the primaries, and may have arisen after metastasis (Supplementary Figs. 14 and 15). Other mutated Cancer Gene Census[17] genes included *MUC16* (five samples, four independent events), which is a common false positive gene encoding the second largest human protein[18], *KMT2C* (five samples, three independent events), *FBLN2* (two independent events), and *ARID1B* (two independent events; Fig. 3). *TERT* promoter mutations, which are frequent non-coding driver events in several cancer types[19,20], were absent, as were notable upstream mutations in other cancer genes (Supplementary Fig. 16). Consistent with other reports, there was thus a lack of obvious driver mutations beyond *CDKN1B*, which was not essential for metastasis.

**Copy number alterations.** Analysis of somatic CNAs gave further support for the phylogenetic relationships inferred from somatic SNVs (Fig. 4a). The metastatic tumor in Patient 1 (C) carried a distinct segmental loss on chr11 found in all metastases (G–K) but not the other intestinal tumors. Similarly, the metastatic tumor in Patient 2 (G) exhibited chromothripsis on chr13 (abundant clustered CNAs that oscillated between two states[21], Supplementary Fig. 17), and this exact complex pattern was mirrored in the corresponding lymph node metastasis (L) (Fig. 4b). Patients 2 and 7 had alterations on chr11 and chr20, respectively, that were present in primary tumors but absent in related metastases and thus presumably occurred after metastasis.

Approximately 70–75% of SI-NETs have been shown to harbor hemizygous loss of chr18[22], and we accordingly observed chr18 loss in nine of the 11 patients and in 32 of the 61 tumor samples. However, phasing of chr18 loss based on germline single nucleotide polymorphisms (SNPs) revealed that tumors deemed unrelated based on SNVs often had undergone loss of different chromosome homologs, while related tumors always showed loss of the same homolog ($P = 1.2 \times 10^{-4}$, binomial test on related monoallelic cases; Fig. 4a, c and Supplementary Fig. 18). Other whole chromosome events (4, 5, 14, and 20) showed similar patterns of concordant/discordant chromosome homolog loss or

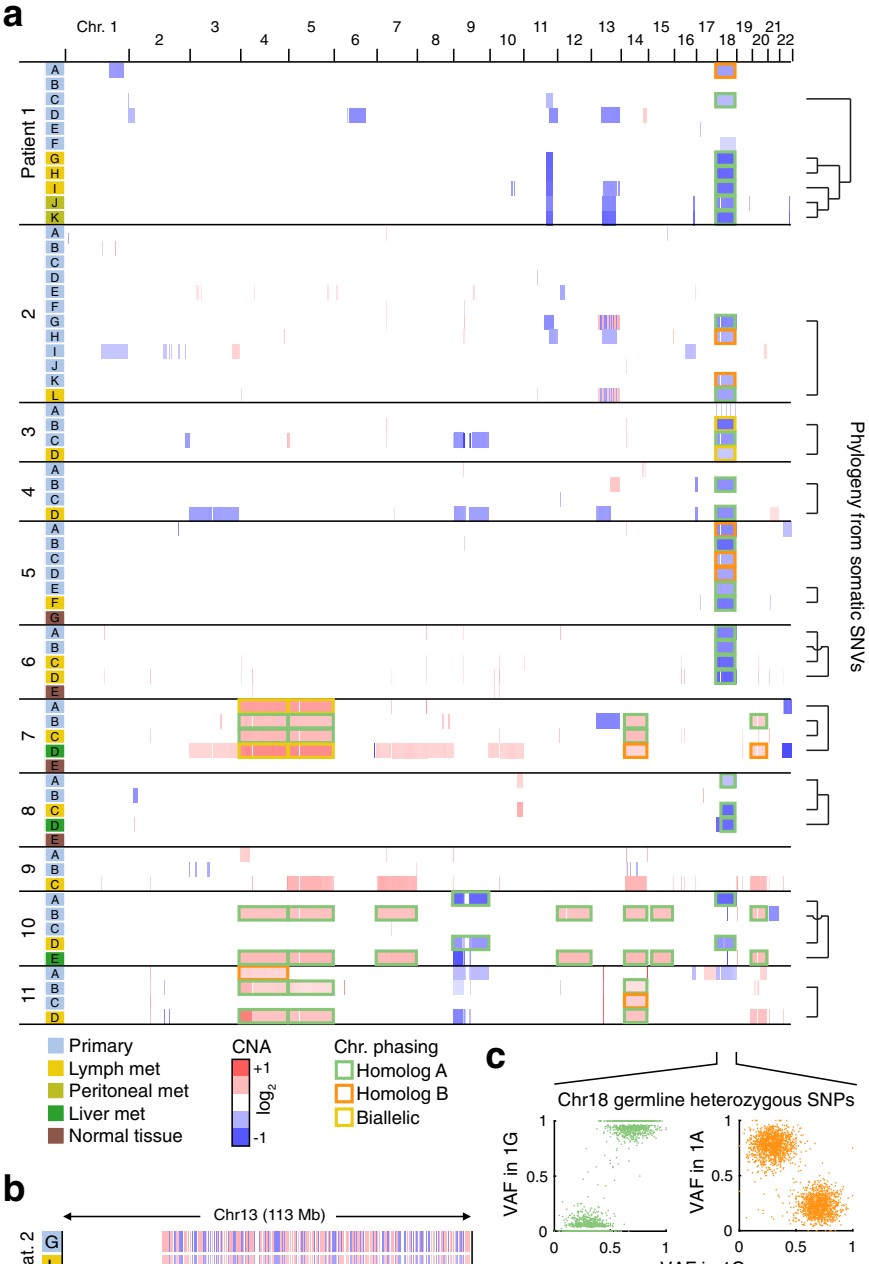

**Fig. 4 Somatic copy number alterations agree with phylogenies inferred from somatic SNVs. a** Somatic CNA profiles for all samples (blue = deletion, red = amplification; segments <25 kb were excluded for visualization purposes). Phylogenetic relationships previously inferred from somatic SNVs (Figs. 1, 2) are indicated to the right. Whole-chromosome losses and gains were phased based on germline SNPs (see Supplementary Fig. 18), with the two chromosome homologs indicated in green and orange and likely biallelic events shown in yellow. **b** Detailed view of chromothripsis on chr13 in Patient 2, present in the metastatic primary (G) and the metastasis (L). **c** Example of phasing of chr18 deletions in Patient 1. Met, metastasis; Chr, chromosome. Source data are provided as a Source data file.

gain in agreement with the established phylogenies (Fig. 4a). Among other recurrent events were segmental deletions on chr11 and chr13, as shown previously[7,23,24]. No distinctive CNAs were shared in ways that contradicted the SNV-based evolutionary trees (Fig. 4a).

Chromothripsis, not previously reported in SI-NET to our knowledge, was further supported by analysis of somatic structural alterations. Of 27 in-frame structural events, none of which were obvious drivers, nine were intrachromosomal alterations on chr13 in Patient 2 that were shared between the primary tumor and the metastasis shown by CNA to harbor chromothripsis on this chromosome (G and L; Supplementary Fig. 19).

## Discussion

We find that in multifocal SI-NET, tumor development is initiated in multiple clonally independent cells, leading to a group of tumors that are unrelated in terms of somatic genetic evolution. Furthermore, we encountered several cases where two independent intestinal tumors had metastasized, each giving rise to a distinct liver or lymph node metastasis. While we found support for the prevailing model of sequential progression via lymph nodes to distant sites (Patient 1), our data also indicated that lymph node and distant metastases may sometimes arise from independent seeding events (Patient 8), as reported previously in colorectal cancer[25]. In the latter case, seeding to distant sites may

still have progressed via undetected, unsampled, or regressed lymph node metastases. Together, our observations suggest that acquisition of metastatic properties is not a rare event in multifocal SI-NET and underscores the importance of complete surgical removal of all intestinal lesions. Previous exome-based analyses of paired single intestinal and metastatic samples have yielded puzzling results with a highly varying degree of genetic overlap and in some cases no overlap at all[7,24], as seen in one patient also in the present study, and our results thus suggest that the relevant primary tumors have not been sampled in such cases.

Multifocal entero-pancreatic carcinomas are typically seen in hereditary cancer syndromes such as MEN-1 and familial adenomatous polyposis (FAP). Cohort studies support that some SI-NETs may have a heritable component[26,27], and multifocality has been suggested to occur preferentially in familial cases[28,29]. However, multifocality is common in all SI-NET[4,6], and key clinical parameters such as survival or age of diagnosis, which is typically lower for hereditary cancer, are similar in patients with or without multifocality[4,6]. The observation that the most recurrent somatic genetic event in SI-NET, hemizygous chr18 loss, can affect both the maternal and the paternal chromosome homologs in the same patient argues against a contribution from a predisposing chr18 germline variant through loss of heterozygosity[12]. While the tumor suppressors DCC and SMAD4 have been nominated as possible drivers of this event through haploinsufficiency, more work is needed to fully understand the role of chr18 loss in SI-NET[30,31]. Furthermore, in stark contrast to germline-induced gastrointestinal tumors, SI-NET only affects a limited intestinal segment.

Given the general lack of established genetic drivers, and in the light of our results showing clonal independence, we suggest that future studies could focus on cancer-priming local factors that may contribute to the emergence of multifocal SI-NET. Such factors could theoretically include embryonic priming of select EC cell lineages through either epigenetic events or early genetic events also present in blood, which would not be detected in this study, or local environmental factors. While such factors remain elusive, it can be noted that SI-NET originates from epithelial enterochrommafin cells (EC-cells) that intrinsically interact with their surroundings by paracrine and endocrine serotonin secretion, synaptic-like connections with the enteric nervous system, and receptor-mediated nutrient sensing of luminal content[32], supporting a possible contribution from changes in the local tumor environment. Further studies directly addressing such alterations might shed further light on the emergence of clonally independent multifocal tumors in SI-NET.

## Methods

**Patients**. Eleven patients who underwent surgery for SI-NET at Sahlgrenska University Hospital, Gothenburg, Sweden, were included in the study. The clinical characteristics of the patients are summarized in Supplementary Table 1. Patients were postoperatively diagnosed with well-differentiated neuroendocrine tumor grade 1–2 of the ileum (WHO 2019). Tumor grade and the stage were based on one primary intestinal tumor and extent of lymph node metastases as standard clinical routine. Tumor samples were collected at surgery and were immediately snap-frozen in liquid nitrogen. A piece of each tumor was formalin-fixed and paraffin-embedded for studies by immunohistochemistry. Blood was collected and used as normal tissue in sequencing. All included participants received in-person information by a designated research nurse as well as written information explaining the purpose of the study. In accordance with the ethical permit, consent was obtained in-person and was documented in writing by a signed document or as a designated entry in the patient's medical journal. The study was approved by the Central ethical review board in Gothenburg (Dnr. 833-18).

**Immunohistochemistry**. Sections of all sampled tumors from 11 patients with SI-NET were subjected to antigen retrieval using EnVision FLEX Target Retrieval Solution (high pH) in a Dako PT-Link. Immunohistochemical staining was performed in a Dako Autostainer Link using EnVision FLEX according to the manufacturer's instructions (DakoCytomation). Histopathological evaluation and

assessment of tumor cell content was performed on hematoxylin and eosin-stained sections. Immunohistochemical staining was performed for chromogranin A, synaptophysin, serotonin and somatostatin receptor 2. These antibodies were used: anti-chromogranin A (MAB319/PHE5; Chemicon; diluted 1:1000), anti-synaptophysin (SY38/M0776; Dako; diluted 1:100), anti-serotonin (H209; Dako; diluted 1:10) and anti-SSTR2 (UMB-1; Abcam; diluted 1:50). All collected tumor samples were reviewed by a board certified surgical pathologist (ON).

**DNA extraction**. For DNA extraction, tumor samples of sufficient size and with a purity above 30% were included. DNA from fresh-frozen biopsies as well as blood from each patient was isolated using the allprep DNA/RNA Mini Kit (Qiagen) according to the manufacturer's protocol.

**Whole genome sequencing**. WGS libraries were constructed for 61 primary and metastatic tumors from the 11 patients. Eleven blood normals and four adjacent tissue normals were additionally sequenced for a total of 76 samples. The prepared libraries were sequenced on an Illumina Novaseq 6000 using 150 bp paired-end reads to an average depth of 36.6 × (29.8 – 45.0).

**Read mapping and somatic variant calling**. Sequencing reads were mapped to the hg19 human reference genome and somatic variant calling of matched tumor-normal specimens was performed using combined outputs from Mutect2 (GATK v4.1.4.0) and VarScan (v2.3.9). The reads were mapped using BWA as part of Sentieon Genomics Tools (bwa-mem v0.7.15.r1140 for patients 1–5 and 0.7.17.r1188 for patients 6–11), which further also performs deduplication, realignment and sorting. Mutect2 was run using default parameters, and the output was fed into the Mutect2's filterMutectCalls tool with a mean median mapping quality score requirement of 10. With VarScan, the somatic tool was used to call variants by default cutoffs and a minimum variant allele frequency of 0.01 from SAMtools (v1.9) pileups of reads with Phred-scaled mapping quality >= 15 and variant base quality >= 20. The VarScan processSomatic tool was then used to get only the somatic SNVs and indels, where minimum support of two reads was required for a variant to be called. The VarScan somaticFilter tool was also used to discard SNVs called within SNP clusters or within 3 bp of somatic and germline indels. Variants were further filtered to have coverage >= 20 reads in the normal, to be present on both strands, and to be absent in the normal. The final set of variants was yielded by intersecting outputs from the two callers, followed by annotation using ANNOVAR (v2019Oct24)[33] and Variant Effect Predictor (VEP)[34]. To minimize false positive calls in phylogenetic analyses, population variants in dbSNP150 (provided by ANNOVAR) and the ENSEMBL variant database (provided by VEP) were removed. Analysis of potential driver mutations and mutational signatures was based on less stringent population variant filtering (dbSNP138), to avoid false negatives and since extensive SNP filtering can skew results from mutational signatures analyses. For high-sensitivity analysis of pairwise shared variants, mutations detected by the standard pipeline as described above in any of the two samples in a pair were whitelisted for detection using the unfiltered VarScan somaticFilter output, allowing more sensitive detection of overlapping subclonal mutations. Intersection and postprocessing of VarScan and Mutect2 calls ware done in MATLAB (r2018a).

**Sample purity and telomere lengths**. Samples were assessed for tumor cell content using PurBayes[35] and telomere lengths were determined using Computel[36] using default settings, both presented in Supplementary Data 1.

**Viral reads and transposition events**. Analysis of WGS reads for viral content, which produced only low counts of expected *Herpesviridae* family reads or reads consistent with plasmid vector contamination, thus giving no support for a role for DNA viruses in SI-NET, was performed using a previously established pipeline[37]. TraFiC-mem was used for detection of somatic mobile element insertions[38], which revealed only a single LINE1 transposition event (in *BCAS3* in sample 5F).

**Phylogenetic analyses**. Phylogenetic analyses were based on SNVs only, as indel calls contained a higher fraction of problematic calls. Maximum parsimony phylogenies were reconstructed and visualized using MEGAX (v10.1.8) with default settings[39]. Bootstrapping was performed using 500 replications.

**Mutational signatures**. The autosomal trinucleotide profile for each tumor was matched against known mutational signatures from COSMIC[16] (v3 release) using the R package deconstructSigs[40] (version 1.9.0). We used default parameters and a maximum of 4 signatures for each sample.

**Copy number analysis and homolog phasing**. Copy number alterations (CNAs) were called using XCAVATOR v2.1[41] in paired/somatic mode and a window size of 2000 bp using the RC option (analysis based on read counts per window). The segmented copy number results were filtered to remove small segments (<25 kb) prior to visualization using IGV. Whole-chromosome alterations were phased to determine what chromosome homolog was deleted based on SNP calls reported by

VarScan somatic ("germline" and "LOH" sets with a required coverage of 20). Only heterozygous SNPs were considered (variant allele frequency ranging from 0.25 to 0.75 in the blood normal). Variant allele frequencies for individual SNPs were compared between a reference sample, typically the metastatic primary tumor in each patient, and other samples of interest by means of scatter plots.

**Structural variant analyses**. Structural variants (SVs) were called based on discordant read pairs and split reads using Manta (v1.5.0)[42]. Genomic coordinates of high-confidence SV calls were annotated using AnnotSV (v2.1)[43]. Of main interest were rearrangements with breakpoints in two genes, which were further investigated to determine if they were on the same strand in the final fusion, and whether the two coding sequences were fused in-frame, thus resulting in a putative valid fusion gene. This was done based on exon and coding sequence coordinates provided by AnnotSV.

**Reporting summary**. Further information on research design is available in the Nature Research Reporting Summary linked to this article.

## Data availability

The WGS sequencing data that support the findings in the article are deposited and available at the European Genome–Phenome Archive (EGA; https://ega-archive.org), which is hosted by the European Bioinformatics Institute (EBI) and the Centre for Genomic Regulation (CRG), through the primary accession code EGAS00001005096. Due to ethical and legal reasons, the data is deposited under controlled access. Data use conditions attached to this EGA dataset limits its use to approved users at a specific institution for a specific health/medical/biomedical project and dictates that useful results should be made available to the wider scientific community. Access requests, which we aim to respond to within two weeks, should be addressed to Erik Elias (erik.elias@gu.se) or Erik Larsson (erik.larsson@gu.se). The study makes use of SNP data from ENSEMBL/VEP (https://grch37.ensembl.org/Homo_sapiens/Tools/VEP; downloaded on Jan 13 2021) and dbSNP (https://www.ncbi.nlm.nih.gov/snp/; v138 and v150). Mutational signatures were downloaded from COSMIC (https://cancer.sanger.ac.uk/signatures/; v3 May 2019). Source data are provided with this paper.

## Code availability

Analysis scripts are available from the authors upon request.

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

## Acknowledgements

The work described here was supported by the Swedish Research Council (E.L.), the Swedish Cancer Society (E.L. and O.N.), the Knut and Alice Wallenberg Foundation (E.L.), the BioCARE National Strategic Research Programme (O.N.), and grants from the Swedish state under the agreement between the Swedish government and the county councils, the ALF agreement (E.E. and O.N.). We wish to thank laboratory assistant Gulay Altiparmak, research nurse Maria Nilsson and surgical coordinator Jenny Oliver for skilled technical assistance, and Kerryn Elliott for critical reading of the manuscript. We further want to thank professor emeritus Bo Wängberg as well as previous and current surgical staff at the Section of Endocrine and Sarcoma Surgery, Department of Surgery, Sahlgrenska University Hospital. Finally, we wish to thank the patients who participated in this study.

## Author contributions

E.E. and E.L. designed the study; E.E., A.A., M.L., S.E.R., O.N., Y.A. and E.L. analyzed data; E.E., A.M. and O.N. acquired clinical materials; E.L. and E.E. drafted the manuscript with contributions from all other authors.

## Funding

## Competing interests

The authors declare no competing interests.
