## [Peer Review File · Nature Communications]

Independent somatic evolution underlies clustered neuroendocrine tumors in the human small intestineREVIEWER COMMENTS

Reviewer #1 (Remarks to the Author):

In their manuscript entitled "Independent somatic evolution underlies clustered neuroendocrine tumors in the human small intestine", Elias et al. analyze multifocal (or I should better say, multicentric) neuroendocrine tumors of the small bowel and their metastases. They find that in all of 11 analyzed patients, the primary tumors represent independent clonal expansions with no shared somatic ancestry. Metastases can easily be assigned to their tumor of origin via WGS, as expected. As previously reported, the mutation and driver burden of the neuroendocrine tumors is minimal.

This manuscript was a real joy to read. The analysis was logical and well-presented, and the paper is exceptionally clearly written. I wish more papers were this clear! All analyses are state-of-the-art from a technical point of view as far as I can tell, and I would be hard pressed to suggest any substantial improvements to the main narrative. If the authors wanted to, they could study subclonal heterogeneity in primaries vs. metastases a bit more, but this is not a requirement. I would be curious if the primaries have more subclonal variants than the metastases (as they should, since they are presumably older). A few plots of the CCF of variants across the different lesions would probably do the trick. I liked Supplementary Figure 6, but it seems like these plots only show the shared mutations, it would be nice to see all of them after scaling for purity. If the authors recorded the size of the primaries (as it seems they did from the supplement), it might also be neat to see if subclonal diversity correlates with lesion size.

I think the authors' findings are very important for pointing the way towards an improved understanding of the etiology of this rare tumor type. In the absence of shared ancestry, what is responsible for the accumulation of these tumors within a particular bowel segment? The authors rightly point to the strong possibility of an environmental factor (once shared ancestry of multiple lesions is excluded as a possibility). This is an important insight and truly fascinating.

In a shameless plug of my own work (sorry!!), I will say that with respect to the independent origins of lymph node and liver metastases (particularly in patient 8), the authors might cite Naxerova et al. Science 2017 for a similar finding in colorectal cancer. I strongly emphasize this is not a necessity though.

Reviewer #2 (Remarks to the Author):

The authors performed whole genome sequencing of multiple samples of primary tumors/metastases from 11 patients diagnosed with multifocal small intestinal neuroendocrine tumor. 61 tumor/metastasis samples were sequenced in total; blood DNA from each patient was used as germline DNA to filter somatic mutations. Additionally, four normal intestinal mucosa from the analyzed cases were sequenced to determine baseline mutational load of non-neoplastic tissue. DNA alterations of primary tumors and matched metastases from each case were used to detect phylogenetic relationships between each sample, showing an independent origin of multiple tumor foci and an independent development of metastases from one or more primary tumors in each patient.

The manuscript covers an interesting and still debated topic presenting a good wealth of evidence. It is well written and easy to follow, and images are produced in a way that clearly conveys the core message. The discussion is agile and focused. While I compliment the authors for their meticulous work, I have a few minor suggestions.

1. Page 5, "DNA extraction". The first sentence is irrelevant here, it will be more useful in the results. You can remove it without changing the message of the paragraph.

2. Page 6, First line: Please briefly explain what is the RC mode of XCAVATOR if the information is relevant or remove the acronym.
3. Page 9, beginning of last paragraph. Please explain a little more (to the non-technical reader) why using the same blood normal sequencing data may increase false shared somatic SNVs.
4. Discussion, page 14 last 4 lines. Regarding chr18, the authors may discuss the topic of possible haploinsufficient genes in chr18, which is further reinforced by their data about the non-importance of which of the two copies of the chromosome is lost.
5. Discussion, page 15 last paragraph. This sentence seems to lack a strong conclusion, the authors should clarify what the intrinsic origin of SiNET from EC cells and their interactions mean to them in this context.
6. Supplementary figures 1 to 5. The images resolution is low so it is difficult to appreciate the histology of each tumor, even at low magnification. I would recommend to upload higher resolution images.

Point-by-point response
Manuscript NCOMMS-20-25690A-Z

“Independent somatic evolution underlies clustered neuroendocrine tumors in the human small intestine”

Dear reviewers,

We want to thank you for your kind comments and for taking the time to evaluate our manuscript.

The reviewer's remarks are shown in black below, while our responses are shown in blue.

Reviewer #1 (Remarks to the Author):

In their manuscript entitled “Independent somatic evolution underlies clustered neuroendocrine tumors in the human small intestine”, Elias et al. analyze multifocal (or I should better say, multicentric) neuroendocrine tumors of the small bowel and their metastases. They find that in all of 11 analyzed patients, the primary tumors represent independent clonal expansions with no shared somatic ancestry. Metastases can easily be assigned to their tumor of origin via WGS, as expected. As previously reported, the mutation and driver burden of the neuroendocrine tumors is minimal.

This manuscript was a real joy to read. The analysis was logical and well-presented, and the paper is exceptionally clearly written. I wish more papers were this clear! All analyses are state-of-the-art from a technical point of view as far as I can tell, and I would be hard pressed to suggest any substantial improvements to the main narrative. If the authors wanted to, they could study subclonal heterogeneity in primaries vs. metastases a bit more, but this is not a requirement. I would be curious if the primaries have more subclonal variants than the metastases (as they should, since they are presumably older). A few plots of the CCF of variants across the different lesions would probably do the trick. I liked Supplementary Figure 6, but it seems like these plots only show the shared mutations, it would be nice to see all of them after scaling for purity. If the authors recorded the size of the primaries (as it seems they did from the supplement), it might also be neat to see if subclonal diversity correlates with lesion size.

We were happy to see the positive remarks from the reviewer. Fig. S6 shows VAF distributions for all mutations (red graphs), but only for tumors that are part of primary-metastasis pairs. We have looked at the complete set of distributions for all samples, but the results were not conclusive: although VAFs are generally higher for metastases, this is not true following scaling for purity as estimated from histology (the PurBayes estimates presented in Table S2 are in

themselves strongly correlated with mean VAF and therefore seems unsuitable in this context). Similarly, preliminary analyses in relation to tumor size (based on patients 1-5) revealed no relationship. Although subclonal diversity may also be indicated by multimodality in these plots, we did not see obvious differences comparing primaries and metastases. We further acknowledge that these analyses are in practice more complicated, as copy number events play a part as well. Given the lack of conclusive results and that these questions, while interesting, are not essential to the main conclusions, we would prefer to leave this out.

I think the authors' findings are very important for pointing the way towards an improved understanding of the etiology of this rare tumor type. In the absence of shared ancestry, what is responsible for the accumulation of these tumors within a particular bowel segment? The authors rightly point to the strong possibility of an environmental factor (once shared ancestry of multiple lesions is excluded as a possibility). This is an important insight and truly fascinating.

We agree and are glad that the reviewer shares our excitement about these findings. As the mechanism remains unknown, we have added a last sentence to the discussion emphasizing the need for further studies in relation to changes in the local tumor environment:

“Further studies directly addressing such alterations might shed further light on the emergence of clonally independent multifocal tumors in SI-NET.”

In a shameless plug of my own work (sorry!!), I will say that with respect to the independent origins of lymph node and liver metastases (particularly in patient 8), the authors might cite Naxerova et al. Science 2017 for a similar finding in colorectal cancer. I strongly emphasize this is not a necessity though.

This is a very relevant study and we are happy to include this reference. The discussion was amended as follows:

“While we found support for the prevailing model of sequential progression via lymph nodes to distant sites (Patient 1), our data also indicated that lymph node and distant metastases may sometimes arise from independent seeding events (Patient 8), as reported previously in colorectal cancer (Naxerova et al., 2017). In the latter case, seeding to distant sites may still have progressed via undetected, unsampled, or regressed lymph node metastases.”

Reviewer #2 (Remarks to the Author):

The authors performed whole genome sequencing of multiple samples of primary tumors/metastases from 11 patients diagnosed with multifocal small intestinal neuroendocrine tumor. 61 tumor/metastasis samples were sequenced in total; blood DNA from each patient was used as germline DNA to filter somatic mutations. Additionally, four normal intestinal mucosa from the analyzed cases were sequenced to determine baseline mutational load of non-neoplastic tissue. DNA alterations of primary tumors and matched metastases from each

case were used to detect phylogenetic relationships between each sample, showing an independent origin of multiple tumor foci and an independent development of metastases from one or more primary tumors in each patient.

The manuscript covers an interesting and still debated topic presenting a good wealth of evidence. It is well written and easy to follow, and images are produced in a way that clearly conveys the core message. The discussion is agile and focused. While I compliment the authors for their meticulous work, I have a few minor suggestions.

We thank the reviewer for these positive comments.

1. Page 5, "DNA extraction". The first sentence is irrelevant here, it will be more useful in the results. You can remove it without changing the message of the paragraph.

It seems this sentence erroneously ended up in the wrong place. We are grateful for having this pointed out. The sentence has been removed (it was already in the results, too).

2. Page 6, First line: Please briefly explain what is the RC mode of XCAVATOR if the information is relevant or remove the acronym.

"RC" refers to the "read counts per window" based mode of operation (as opposed to read depth-based), which has now been clarified. Additionally, we added information that we are using XCAVATOR's paired/somatic mode.

3. Page 9, beginning of last paragraph. Please explain a little more (to the non-technical reader) why using the same blood normal sequencing data may increase false shared somatic SNVs.

We agree that some clarification is useful as this may not be obvious to everyone. False positives in somatic calling often stem from failure to detect a variant in the normal data (germline variants or false variants e.g. due to alignment issues). These errors are always present to some extent (e.g. when coverage is low, there is a small probability of simply not sampling a heterozygous germline SNP). The problem is that, when several samples share the same normal data, these errors become systematic. The sentence was reworded:

"False shared somatic SNVs may arise, for example, due to failure to detect germline SNPs at specific positions in the blood normal, since this data is common to all samples in a patient."

4. Discussion, page 14 last 4 lines. Regarding chr18, the authors may discuss the topic of possible haploinsufficient genes in chr18, which is further reinforced by their data about the non-importance of which of the two copies of the chromosome is lost.

We agree that this is relevant. A sentence about this was added that includes references to two relevant studies that nominate putative drivers of chr18 through haploinsufficiency, including one that we published this year.

5. Discussion, page 15 last paragraph. This sentence seems to lack a strong conclusion, the authors should clarify what the intrinsic origin of SiNET from EC cells and their interactions mean to them in this context.

We simply mean to say that these basic properties of the cell of origin, the EC cell, support a possible role for local microenvironmental changes in development of SI-NET, although we want to avoid speculating further than this. We added a concluding remark to this sentence, to clarify this:

“While such factors remain elusive, it can be noted that SI-NET originates from epithelial enterochromaffin cells (EC-cells) that intrinsically interact with their surroundings by paracrine and endocrine serotonin secretion, synaptic-like connections with the enteric nervous system, and receptor-mediated nutrient sensing of luminal content (Bellono et al., 2017), supporting a possible contribution from changes in the local tumor environment.”

6. Supplementary figures 1 to 5. The images resolution is low so it is difficult to appreciate the histology of each tumor, even at low magnification. I would recommend to upload higher resolution images.

Thank you for notifying us about this, which in part was due to automatic downsampling of images in word. The resolution in the PDF has now been increased to what we hope is a satisfactory level.

REVIEWERS' COMMENTS

Reviewer #1 (Remarks to the Author):

Many thanks, I am satisfied with the authors' replies. Congratulations on a nice paper.